# Optimization Method of Tool Parameters and Cutting Parameters Considering Dynamic Change of Performance Indicators

**DOI:** 10.3390/ma14206181

**Published:** 2021-10-18

**Authors:** Daxun Yue, Anshan Zhang, Caixu Yue, Xianli Liu, Mingxing Li, Desheng Hu

**Affiliations:** Key Laboratory of Advanced Manufacturing and Intelligent Technology, Ministry of Education, Harbin University of Science and Technology, Harbin 150080, China; 1920100002@stu.hrbust.edu.cn (D.Y.); yuecaixu@hrbust.edu.cn (C.Y.); Xianli.liu@hrbust.edu.cn (X.L.); 1910100005@stu.hrbust.edu.cn (M.L.); 1920110094@stu.hrbust.edu.cn (D.H.)

**Keywords:** cutting performance, matching combination, dynamic change process of performance indicator, dynamic evaluation method, comprehensive evaluation

## Abstract

In the process of metal cutting, the cutting performance of cutting tools varies with different parameter combinations, so the results of the performance indicators studied are also different. So in order to achieve the best performance indicator it is necessary to get the best parameter matching combination. In addition, in the process of metal cutting, the value of the performance index is different at each stage of the processing process. In order to consider the cutting process more comprehensively, it is necessary to use a comprehensive evaluation method that can evaluate the dynamic process of performance indicators. This paper uses a dynamic evaluation method that considers the dynamic change of performance indicators in each stage of the cutting process to comprehensively evaluate the tool parameters and cutting parameters at each level. For the purpose of high processing efficiency and long tool life, tool wear rate and material removal rate are used as performance indicators. In the case of specified rake angle, cutting speed and cutting width, titanium alloy is studied by end milling cutter side milling. The tool parameters and cutting parameters in milling process are optimized by using a dynamic comprehensive evaluation method based on gain horizontal excitation. Finally, the parameter matching combination that can make the performance indicator reach the best is obtained. The results show that when the rake angle is 8°, the cutting speed is 37.68 m/min, and the cutting width is 0.2 mm, the tool wear rate and material removal rate are the best when the clearance angle is 9°, the helix angle is 30°, the feed per tooth is 0.15 mm/z, and the cutting depth is 2.5 mm.

## 1. Introduction

Metal cutting is one of the most widely used metal parts manufacturing methods [1]. However, in the process of machining, the cutting performance of the tool is different under different parameter combinations, and the required performance indicator results are also different. Therefore, in order to make the performance indicator reach the optimal value, it is necessary to study the matching between each parameter in the cutting process, and get the optimal parameter matching combination of each performance indicator.

At present, many scholars have studied the optimal matching combination among parameters. Zhang et al. [2] conducted the orthogonal test of high speed milling of aviation aluminum alloy with end mills, and obtained the optimal combination of milling parameters through range analysis. Kubilay et al. [3] taking the turning of Ti6Al4V with indexable turning tool as the research object. The average roughness height, maximum roughness height, and material removal rate were taken as the performance indicators. The parameters are cutting speed, feed speed, and cutting depth. The surface response function of three performance indicators was obtained by using the surface response method, and the optimal parameter combination was obtained by multi-objective optimization algorithm. Tamal et al. [4] proposes a bayesian regularization neural network for agents and the beetle antenna search algorithm to optimize the algorithm of data driven auxiliary agent optimization algorithm, the algorithm we get the material removal rate, surface roughness and cutting force on cutter diameter, spindle speed, feed speed, and cutting depth function and to get the optimized parameters combination. Juan et al. [5] used the orthogonal cutting simulation method to obtain the relevant performance indicator values as optimization parameters, and proposed a multi-objective particle swarm optimization algorithm to optimize the performance indicator values. Zhang et al. [6] used the Markov chain Monte Carlo method to solve the reliability model of tool life, and then used the multi-objective optimization algorithm combining grey correlation analysis, radial basis neural network and particle swarm optimization algorithm to search for the optimal processing parameters of the whole blade-disc tunnel processing. Mohammed et al. [7] took cutting force and surface roughness as performance indicators and combined gray correlation method (GRA) and expectation function analysis (DFA) to optimize the milling parameters in the milling process of epoxy glass fiber to obtain the best combination of milling parameters. Fang et al. [8] respectively established the correlation function model of the above performance indicators on milling times and milling parameters for the energy consumption, processing cost, and processing time of CNC machine tools in the process of multi-pass milling. On this basis, an improved adaptive simulated annealing particle swarm optimization algorithm was proposed to solve the optimal solution of milling parameters, and the optimal combination of milling parameters was obtained. Vimal et al. [9] took surface roughness, tool wear, and cutting force as performance indicators and used grey-fuzzy evaluation method to optimize the cutting speed, feed speed, and cutting depth in the turning process of glass fiber reinforced plastics, and obtained the optimal parameter combination of cutting speed, feed speed, and cutting depth. Viswanathan et al. [10] took PVD coated carbide turning tools dry turning magnesium alloy as the research object, taking cutting speed, feed per revolution, and cutting depth as the optimized parameters, and taking cutting force, material removal rate, flank face wear, and surface roughness as the performance indicators, then carried out a parameter optimization test. Principal component analysis (PCA) and grey correlation analysis (GRA) were used to optimize the parameters and get the best combination of parameters. Suresh et al. [11] taking surface roughness and material removal rate as performance indicators, the grey-fuzzy comprehensive evaluation method was used to optimize the cutting speed, feed per revolution, and mass fraction of SiC-Gr in the turning process of Al-SiC-Gr composites. Gnanavelbabu et al. [12] controlled turning of aluminum-based boron carbide composites numerically, and the cutting force, tool wear, and other performance indicators in the turning process are measured. The optimum parameter combination among spindle speed, feed per revolution, cutting depth, and B_4_C mass fraction was obtained by grey-fuzzy comprehensive evaluation method.

Metal cutting is a dynamic process, so the performance indicator values are constantly changing with the movement of the cutting tool in the cutting process. Based on the above reference studies, it can be seen that the research methods of parameter matching include range analysis method, optimization algorithm, and evaluation method to obtain the best parameter level combination, but these methods do not consider the data dynamic change process of performance indicator in different stages of cutting process. Therefore, it is important to use an optimization method that takes into account the data of all performance indicators and the data dynamic change process of performance indicators in different stages.

Ti6Al4V is a difficult material to process and is widely used in many industrial fields because of its good thermodynamic properties [13,14]. However, when cutting titanium alloy, there will be serious friction between the tool and the workpiece, so there is a large cutting force and high temperature in the cutting contact area, these problems lead to accelerated tool wear speed, tool life is low, and ultimately lead to low processing efficiency, affecting the application of Ti6Al4V [15,16]. In addition, milling is one of the most commonly used machining processes because of its ability to produce complex geometric shapes [17,18]. Therefore, this paper chooses milling as the processing method to be studied. In this paper, for the purpose of high efficiency and long tool life, this paper takes tool wear rate and material removal rate as performance indicators, and titanium alloy is milled by end milling cutter side as the research object. According to the dynamic change of performance indicator, the dynamic evaluation method based on gain level excitation in reference [19] was used to comprehensively evaluate each level of tool parameters and cutting parameters, and finally the optimal level of each parameter on tool wear rate and material removal rate was optimized.

## 2. Establishment and Verification of Finite Element Simulation of Milling Process

Finite element method has the characteristics of a simple and clear physical concept, is easy to grasp, has a simple description, is easy to popularize, is a superior method, and has a wide application range. In this paper, the tool wear rate and material removal rate are obtained by finite element method and analytical method under different tool parameter and cutting parameter combinations at different stages.

### 2.1. Establishment of Finite Element Simulation Model for Milling Process

#### 2.1.1. Finite Element Simulation Model Establishment Process

Finite element simulation is an effective tool to evaluate metal cutting process. In recent years, many researchers have carried out finite element simulation analysis on metal cutting process. Finite element method can be used for chip forming simulation, cutting force simulation, wear simulation, etc [20]. At present, many software have been used in metal cutting process simulation research [21]. Deform-3D (Ohio, USA) is a powerful software for simulation and analysis of metal cutting processes [22]. Finite element simulation of cutting process using Deform-3D has advantages such as reducing workpiece cost and machine energy consumption [23]. Deform-3D simulation system is composed of three main modules, which are the pre-processing module, simulation setting module, and post-processing module [24]. According to the above three modules, Figure 1 shows the process of establishing the cutting simulation model of Deform-3D software.

#### 2.1.2. Material Constitutive Model

The constitutive equation of titanium alloy includes stress, strain, temperature, and other parameters [25]. Johnson–Cook material model is simple in form and widely used. It is an ideal model of elastic-plastic strengthening. The model is suitable for the temperature range from room temperature to the melting point of the material [26]. So, the J-C constitutive reinforcement model was selected. The expression form of J-C constitutive model equation is shown in Equation (1).
(1)$$\bar{\sigma }=\left[A+B{\left(\bar{\varepsilon }\right)}^{n}\right]\cdot \left[1+C\ln\frac{\bar{\varepsilon }}{{\bar{\varepsilon }}_{0}}\right]\cdot \left[1-{\left(\frac{T-T_{r}}{T_{m}-T_{r}}\right)}^{m}\right]$$
where *σ*, *ε* and *ε*_0_ are equivalent flow stress, equivalent plastic strain rate, and reference plastic strain rate, respectively. *T*, *T_r_*, and *T_m_* denote the absolute temperature, ambient temperature and melting temperature of the workpiece material, respectively. *A*, *B*, *C*, *m*, and *n* are the yield strength, hardening modulus, strain rate sensitivity coefficient, heat softening coefficient, and strain hardening index, respectively. J-C parameters of the Ti6Al4V constitutive model are presented in Table 1.

#### 2.1.3. Material Parameters

In the pre-processing of finite element simulation, the material attributes of the tool and the workpiece need to be set, and the setting of material parameters plays a crucial role in the accuracy of finite element simulation. The cutter material is carbide, model is YG6, and the tool is not coated. The workpiece material is Ti6Al4V. Table 2 shows the material parameters of tool material YG6 and workpiece material Ti6Al4V.

#### 2.1.4. D Model Establishment, Import and Grid Division

UG is used to conduct 3D modeling of the tool and workpiece, Figure 2a is the model of integral end milling cutter. Figure 2b is the model of square block workpiece, the length is 100 mm. Table 3 shows the key parameters of the integral end milling cutter. In order to speed up the finite element simulation, the 3D model of tool and workpiece is simplified. Because the research object is the end milling cutter side milling block titanium alloy, the main study is of the integral end milling cutter side edge, cut end milling cutter only with a part of the edge, and cut block workpiece in the cutting area of the nearest part of the workpiece. The height of the cut part of the workpiece is determined by the cutting depth. After the model is assembled in UG, all parts are exported in the format of stl file. After all parts are imported into DEFORM-3D software, the original assembly form can still be obtained. Figure 2c shows the tool model and workpiece model after simplifying and importing Deform-3D software.

After importing the 3D model into the DEFORM-3D software, the tool model and the workpiece model are set as rigid body and plastic body respectively. After that, the parts are meshed, and the mesh number of the tool model is 30,000. The number of mesh of the workpiece varies according to the cutting depth. In Figure 2c, the thickness of the workpiece is 2 mm, so the number of mesh of the workpiece is 20,000. When the cutting depth is 3 mm, the mesh number of workpiece is 30,000. When the workpiece is meshing, the workpiece is divided into cutting zone and non-cutting zone. The mesh size of the cutting zone is 0.2, and the mesh size of the non-cutting zone is 4. The resulting figure after mesh division is shown in Figure 2d.

#### 2.1.5. Setting of Tool Wear Model in Finite Element Simulation Software

In this paper, the tool wear rate is selected as one of the performance indicators. Therefore, it is very important to set the tool wear rate in the pre-processing of finite element simulation software. The tool wear rate in Deform-3D is set in the “Tool Wear” window of the “Inter-Object” window. Usui wear model and other wear models can be set in this column. Figure 3 shows the tool wear model setting window.

### 2.2. Simulation Parameter Selection and Performance Indicator Setting

In this paper, the tool wear rate and material removal rate as performance indicators. In this chapter, the data values of tool wear rate and material removal rate are obtained through finite element simulation and analytical method.

#### 2.2.1. Simulation Parameter Selection

According to the comprehensive evaluation study on the importance of parameters to performance indicators in the process of titanium alloy side milling by end milling cutter in the early stage, the clearance angle and helix angle are selected as important tool parameters, and the feed per tooth and cutting depth are important cutting parameters. In this paper, the clearance angle, helix Angle, feed per tooth, and cutting depth are the research objects, and the optimal parameter combination is studied. The values of other parameters not involved in the study are shown in Table 4.

According to the selected four parameters with the greatest importance, they are taken as the horizontal factor of orthogonal test to carry out test planning. Without considering the interaction between parameters, SPSS software was used to design the orthogonal test of 4 factors and 5 levels, and the parameter level table of L_25_(5^4^) orthogonal test was obtained as shown in Table 5 and Table 6 was the orthogonal simulation test table corresponding to Table 5.

#### 2.2.2. Setting Performance Indicators

One of the performance indicators in this paper is material removal rate, and the research object is milling block titanium alloy with end milling cutter. Therefore, the formula of material removal rate V within a certain time is shown in Equation (2).
(2)$$V=v_{f}\cdot a_{p}\cdot a_{e}=\frac{50\cdot v_{c}\cdot f_{z}\cdot z}{3\cdot \pi \cdot d}\cdot a_{p}\cdot a_{e}$$
where *v_f_* is feed speed, *v_c_* is cutting speed, *f_z_* is feed per tooth, *z* is the number of teeth, *d* is cutter diameter, *a_p_* is cutting depth, *a_e_* is cutting width.

Many scholars have done a lot of research on extending tool life by controlling tool wear rate [29], so tool wear rate is chosen as the performance indicator. In the metal cutting process, tool wear is defined as the material loss or deformation of the contact surface caused by friction between the cutting tool and the workpiece. Generally, the tool wear can be mainly divided into adhesive wear, abrasive wear, and so on [30]. The tool wear rate is chosen as one of the performance indicators, so it is extremely important to choose the right wear model. In the process of carbide cutting tools processing Ti6Al4V, the tool wear process under the action of cutting force and cutting heat is very complicated. But adhesive wear will occur regardless of the temperature [31]. Therefore, the adhesive wear model is selected as the research object for FEM, the adhesive wear rate was calculated using the adhesive wear model of Usui et al. [32], its expression is shown in Equation (3).
(3)$$\frac{dW_{\mathrm{Adhesion}\;\mathrm{wear}}}{dt}=A_{w}\cdot \sigma_{n}\cdot v_{c}\cdot e^{\left(\frac{-B_{w}}{273+T}\right)}$$
where *σ_n_* is the positive pressure, *v_c_* is the chip slip velocity, *T* is the celsius temperature, and *A_w_* and *B_w_* are the wear characteristic constants, which can be obtained by tool wear test. According to reference [33] and reference [34], *A_w_* = 0.0004, *B_w_* = 7000, in which the values of *A_w_* and *B_w_* need to be input into Deform-3D software.

### 2.3. Finite Element Simulation Results

After Deform-3D post-processing, the tool wear rate when machining to 10 mm was obtained. The simulation results are shown in Figure 4, and the dynamic change diagram of milling simulation is shown in Figure 5. Among them, the cutting length of 10 mm is divided into 4 sections, respectively marked *l*_1_ phase, *l*_2_ phase, *l*_3_ phase, *l*_4_ phase. According to Equation (2), the material removal rate of titanium alloy processed by end milling cutter is obtained. Table 7 shows the data table of tool wear rate and material removal rate under different combinations of tool parameters and cutting parameters in each stage.

## 3. Dynamic Evaluation Method Based on Gain Horizontal Excitation

In different stages under different cutting tool parameters and cutting parameters combination of tool wear rate and material removal rate for data analysis, using the dynamic evaluation method based on gain level motivation for evaluating the various levels, the first parameter is selected by evaluating the parameters of the numerical biggest level, and so on, to get optimal levels of other parameters, eventually, become the best parameter combination.

### 3.1. Dynamic Evaluation Method Based on Gain Level Excitation

The dynamic comprehensive evaluation method based on gain horizontal excitation is a comprehensive evaluation method that considers the dynamic numerical changes of performance indicators at different stages. This method is based on the gain of each evaluated object in different periods, and then calculates the final total dynamic comprehensive evaluation value of each evaluated object through this dynamic evaluation method. Among them, gain represents the change of comprehensive evaluation value in the upper and lower stages. Level represents the comprehensive evaluation value of each stage; Motivation represents the guidance from one stage to the next. Figure 6 shows the flow chart of dynamic comprehensive evaluation method based on gain horizontal excitation.

As shown in Figure 7, the figure shows that the values of performance indicators of the evaluated object are different in different time periods. The dynamic comprehensive evaluation method not only takes into account all performance indicators, but also takes into account the characteristics that performance indicators may have different values in different stages. In this method, all performance indicators of all evaluated objects are comprehensively evaluated at time *t*_1_ to obtain the comprehensive evaluation value of each evaluated object at that time, and the same method is applied at other times. Then, the evaluation values of each evaluation object at different times are unified together, and the dynamic evaluation values of each evaluated object are obtained according to the dynamic evaluation method.

### 3.2. Comprehensive Evaluation of Each Stage Based on Grey-Fuzzy Analytic Hierarchy Process

The purpose of this paper is to optimize the parameter level, so the sequential stereoscopic data table of each level of the studied parameter is shown in Table 8. There are *n* evaluated objects, that is, each parameter has *n* levels, *s_i_* is the *i*th level of this parameter, m performance indicators, *x_ij_*(*t_k_*) is the *i*th (*i* = *1*, 2, …, *n*) evaluated objects at *t_k_* (*k* = 1, 2, …, *T*) about the indicator *x_j_* (*j* = 1, 2, …, *m*) observed value.

In the process of metal cutting, there exist the influence of parameters on the performance indicator, and there are also the mutual influence between the performance indicator, and the influence degree of these influences is unknown, so the metal cutting process is a fuzzy process. In order to save cost, the number of experiments and data amount in scientific research are limited. Therefore, in order to solve the problem of fuzziness in cutting process and insufficient data due to the limited number of tests, fuzzy theory, and grey theory are needed.

Grey fuzzy evaluation method is an evaluation method that has the advantages of both grey theory and fuzzy theory [35]. The combination of analytic hierarchy process and grey fuzzy evaluation method can better solve the problem of multi-objective comprehensive evaluation. This method is used to comprehensively evaluate the performance indicator data of different evaluated objects at each time to obtain the comprehensive evaluation value of each evaluated object at each time, which is finally combined into the comprehensive evaluation matrix *Y*, whose expression form is shown in Section 3.3. The calculation formula and related parameters of analytic hierarchy process and grey fuzzy evaluation method can be obtained from reference [36] and reference [37]. Through analytic hierarchy process and grey fuzzy evaluation, the comprehensive evaluation matrix *B_k_* of the evaluated object at the *t_k_* moment is obtained, and its expression is shown in Equation (4).
(4)$$B_{k}=\left[\begin{array}{cccc} y_{1}\left(t_{k}\right) & y_{2}\left(t_{k}\right) & \ldots & y_{n}\left(t_{k}\right) \end{array}\right]$$
where, *y_i_*(*t_k_*) (*y_i_*(*t_k_*) ∈ [0,1]) is the comprehensive evaluation value of the ith evaluated object at the *t_k_* moment.

### 3.3. Parameter Level Optimization Based on Dynamic Evaluation Method

After the evaluation matrix of each stage is obtained, the dynamic comprehensive evaluation of the evaluated object is carried out according to the dynamic evaluation method in reference [19], and the dynamic comprehensive evaluation value of the evaluated object is finally obtained.

The comprehensive evaluation matrix *B_k_* of the evaluated object at the *t_k_* moment is obtained through the grey-fuzzy analytic hierarchy process, and all the comprehensive evaluation matrix *B_k_* is combined into the comprehensive evaluation matrix *Y*, as shown in Equation (5).
(5)$$Y=\left[\begin{array}{cccc} y_{1}\left(t_{1}\right) & y_{1}\left(t_{2}\right) & \cdots & y_{1}\left(t_{T}\right)\\ y_{2}\left(t_{1}\right) & y_{2}\left(t_{2}\right) & \cdots & y_{2}\left(t_{T}\right)\\ \vdots & \vdots & \vdots & \vdots \\ y_{n}\left(t_{1}\right) & y_{n}\left(t_{2}\right) & \cdots & y_{n}\left(t_{T}\right) \end{array}\right]$$
where *y_i_*(*t_k_*) is the comprehensive evaluation value of the *i*th evaluated object at the *t_k_* moment.

**Definition** **1.**
*Mean maximum gain, mean minimum gain and mean gain of η^max^, η^min^, and*

$\bar{\eta }$

*evaluated object, respectively. Its calculation formula is shown in Equation (6).*



(6)
$$\begin{array}{l} \eta^{\max}=\underset{i}{\max}\left(\frac{1}{T-1}\overset{T-1}{\underset{k=1}{\Sigma }}\left(y_{i}\left(t_{k+1}\right)-y_{i}\left(t_{k}\right)\right)\right)\\ \eta^{\min}=\underset{i}{\min}\left(\frac{1}{T-1}\overset{T-1}{\underset{k=1}{\Sigma }}\left(y_{i}\left(t_{k+1}\right)-y_{i}\left(t_{k}\right)\right)\right)\\ \bar{\eta }=\frac{1}{n\left(T-1\right)}\overset{n}{\underset{i=1}{\Sigma }}\overset{T-1}{\underset{k=1}{\Sigma }}\left(y_{i}\left(t_{k+1}\right)-y_{i}\left(t_{k}\right)\right) \end{array}$$


**Definition** **2.**
*Respectively, η^+^ and η^−^ are the good and bad gain levels of the evaluated object, and their calculation formula is shown in Equation (7).*

(7)
$$\begin{array}{l} \eta^{+}=\bar{\eta }+k^{+}\left(\eta^{\max}-\bar{\eta }\right)\\ \eta^{-}=\bar{\eta }-k^{-}\left(\bar{\eta }-\eta^{\min}\right) \end{array}$$

*where k^+^ and k^−^ are corresponding floating coefficients, k^+^ and k^−^ ∈ (0,1]. Floating coefficients k^+^ and k^−^ are used to describe the decision maker’s psychological expectation of the overall development of the evaluated object.*


After the good and bad gain levels *η*^+^ and *η*^−^ are obtained, they are substituted into the following Equation (8),
(8)$$\begin{array}{l} \eta^{+}=y_{i}^{+}\left(t_{k}\right)-y_{i}\left(t_{k-1}\right),(k=2,3,\ldots ,T)\\ \eta^{-}=y_{i}^{-}\left(t_{k}\right)-y_{i}\left(t_{k-1}\right),(k=2,3,\ldots ,T) \end{array}$$

At this point, the excellent and bad excitation points *y_i_*^+^(*t_k_*) and *y_i_*^−^(*t_k_*) of the *i*th evaluated object at the *t_k_* moment are obtained by Equation (8).

The excellent and bad excitation points *y_i_*^+^(*t_k_*) and *y_i_*^−^(*t_k_*) are substituted into Equation (9).
(9)$$\begin{array}{l} \upsilon_{i}^{+}\left(t_{k}\right)=y_{i}\left(t_{k}\right)-y_{i}^{+}\left(t_{k}\right),y_{i}\left(t_{k}\right)>y_{i}^{+}\left(t_{k}\right)\\ \upsilon_{i}^{-}\left(t_{k}\right)=y_{i}^{-}\left(t_{k}\right)-y_{i}\left(t_{k}\right),y_{i}^{-}\left(t_{k}\right)>y_{i}\left(t_{k}\right) \end{array}$$
where *υ_i_*^+^(*t_k_*) and *υ_i_*^−^(*t_k_*) are the excellent and bad excitation quantities obtained by the ith evaluated object at the *t_k_* stage, respectively. In addition, in the case outside the value range, the superior and inferior excitation quantities are 0. *υ_i_*^+^(*t_k_*) = *υ_i_*^−^(*t_k_*) = 0 is set at the initial *t_k_* stage without any excitation. Figure 8 is the geometric visual representation of the excellent and bad excitation points. In Figure 8, *t_k_*, *t*_*k*+1_ and *t*_*k*+2_ stages respectively represent the three situations in which the evaluated object obtains the excellent excitation, does not obtain the bad excitation and obtains the good excitation.

After obtaining the excellent and bad incentive points, the dynamic comprehensive evaluation value should also consider appropriate rewards and punishments for the parts above and below the excellent and bad incentive points. Let *z_i_*(*t_k_*) be the dynamic comprehensive evaluation value of the *i*th evaluated object in the *t_k_* stage, then the calculation formula of *z_i_*(*t_k_*) is shown in Equation (10).
(10)$$z_{i}\left(t_{k}\right)=h^{+}\upsilon_{i}^{+}\left(t_{k}\right)+y_{i}\left(t_{k}\right)-h^{-}\upsilon_{i}^{-}\left(t_{k}\right)$$
where *h*^+^, *h*^−^(*h*^+^, *h*^−^ > 0) are superior and inferior excitation factors respectively; *h*^+^*υ_i_*^+^(*t_k_*) and *h*^−^*υ_i_*^−^(*t_k_*) are the optimal and the inferior excitation values respectively. In addition, according to Equation (9), any *t_k_*(*k* = 1, 2, …, *T*) in the moment, *υ_i_*^+^(*t_k_*) × *υ_i_*^−^(*t_k_*) = 0, that is, *h*^+^*υ_i_*^+^(*t_k_*), *h*^−^*υ_i_*^−^(*t_k_*) cannot be obtained at the same time.

The proportion rule of the total amount of incentives. For the *n* evaluated objects, Equation (11) shows that the total amount of good and bad excitation is proportional.
(11)$$r=\frac{h^{+}\overset{n}{\underset{i=1}{\Sigma }}\overset{T}{\underset{k=1}{\Sigma }}\upsilon_{i}^{+}\left(t_{k}\right)}{h^{-}\overset{n}{\underset{i=1}{\Sigma }}\overset{T}{\underset{k=1}{\Sigma }}\upsilon_{i}^{-}\left(t_{k}\right)}$$
where *r* (*r* ∈ *R*^+^) is the proportion relation between the total amount of excellent incentives and the total amount of bad incentives, which is a reflection of the decision intention of the evaluator. When *r* > 1, indicates that the total amount of excellent excitation is greater than the total amount of bad excitation; When *r* < 1, indicates that the total amount of excellent excitation is less than the total amount of bad excitation; When *r* = 1, it means that the total amount of excellent excitation is equal to the total amount of bad excitation.

Moderate incentive rules. According to Equation (12), the sum of superior and inferior excitation factors *h*^+^ and *h*^−^ is 1.
(12)$$h^{+}+h^{-}=1$$

When the ratio *r* between the total amount of optimal excitation and the total amount of inferior excitation is determined, the values of *h*^+^ and *h*^−^ can be obtained through Equations (11) and (12).

The dynamic comprehensive evaluation value *z_i_*(*t_k_*) of the ith evaluated object at the *t_k_* stage is obtained through the above steps. Then the total dynamic comprehensive evaluation value *z_i_* of the ith evaluated object at all times is shown in Equation (13).
(13)$$z_{i}=\overset{T}{\underset{k=1}{\Sigma }}\tau_{k}z_{i}\left(t_{k}\right)$$
where *τ_k_* is the time factor, {*τ_k_*} is usually taken as a series of increasing type. If there is no specific requirement and time preference can be ignored, *τ_k_* = 1.

### 3.4. Comprehensive Evaluation of Parameter Level in Each Stage

Firstly, the comprehensive evaluation of each level of the clearance angle was carried out, and the grey fuzzy analytic hierarchy process in reference [36] and reference [37] was used to carry out the comprehensive evaluation, and the comprehensive evaluation value of each level of the clearance angle in each stage was obtained.

Use analytic hierarchy process to get the weight of each performance indicator. According to the requirement, the indicator set is {*x*_1_, *x*_2_}. Among them, *x*_1_ is tool wear rate, and *x*_2_ is material removal rate. First of all, weight distribution was carried out for each performance indicator. In order to highlight the importance of tool life, it was necessary to increase the weight value of tool wear rate, as shown in Table 9. Then, according to reference [36], the weight value of tool wear rate is 0.6, the weight value of material removal rate is 0.4, and the indicator weight matrix *P* = [0.6, 0.4].

According to the weight matrix of performance indicators and the grey fuzzy evaluation method in reference [36], the comprehensive evaluation of the evaluated object in each stage is carried out. As the object is evaluated for each level of parameters. Therefore, all levels of the clearance angle are evaluated comprehensively. According to the performance indicators selected in this paper, phase *l_k_* was used instead of moment *t_k_* to represent the dynamic nodes in the cutting process. The values shown in Table 10 are the average values of the sum of the performance indicator values of the clearance angle at the same level obtained in Table 7 at different stages. And according to the weight matrix of performance indicators and the grey-fuzzy evaluation method, the comprehensive evaluation matrix *B*_1_, *B*_2_, *B*_3_, and *B*_4_ of four stages are obtained, among which the coefficients required by the grey-fuzzy evaluation method can be obtained from reference [37].
$$\begin{array}{l} B_{1}=\left[\begin{array}{ccccc} 0.77701 & 0.88889 & 0.75000 & 0.90000 & 0.57008 \end{array}\right]\\ B_{2}=\left[\begin{array}{ccccc} 0.89557 & 0.78564 & 0.75000 & 0.90000 & 0.61180 \end{array}\right]\\ B_{3}=\left[\begin{array}{ccccc} 0.87574 & 1.00000 & 0.43924 & 0.50000 & 0.38385 \end{array}\right]\\ B_{4}=\left[\begin{array}{ccccc} 0.82976 & 0.91783 & 0.69123 & 0.90000 & 0.33333 \end{array}\right] \end{array}$$

### 3.5. Dynamic Evaluation of Parameter Level

After obtaining the comprehensive evaluation value of each level of the clearance angle in each stage, the dynamic evaluation method is used to carry out comprehensive evaluation, and the final comprehensive evaluation value of each level of the clearance angle is obtained.

The comprehensive evaluation matrices *B*_1_, *B*_2_, *B*_3_, and *B*_4_ of each stage obtained in 3.4 were transformed and combined to obtain the comprehensive evaluation matrix *Y* of five levels about the clearance angle.
$$Y=\left[\begin{array}{cccc} 0.77701 & 0.89557 & 0.87574 & 0.80976\\ 0.88889 & 0.78564 & 1.00000 & 0.91783\\ 0.75000 & 0.75000 & 0.43924 & 0.69123\\ 0.90000 & 0.90000 & 0.50000 & 0.90000\\ 0.57008 & 0.61180 & 0.38385 & 0.33333 \end{array}\right]$$

According to Equation (6), the average maximum gain *η*^max^ = 0.01092, average minimum gain *η*^min^ = −0.07892, average gain $\bar{\eta }$ = −0.015588 were obtained for the comprehensive evaluation matrix *Y*.

Taking the floating coefficients *k*^+^ and *k*^−^ as 0.3, the optimal gain level *η*^+^ = −0.00764 and the inferior gain level *η*^−^ = −0.03459 of the comprehensive evaluation matrix *Y* were obtained according to Equation (7).

According to Equation (8), the optimal excitation points *y_i_*^+^(*t_k_*) and the inferior excitation points *y_i_*^−^(*t_k_*) of each level of the clearance angle in Table 11 at different stages were obtained.

According to Equation (9), the optimal excitation quantities *υ_i_*^+^(*l_k_*) and the inferior excitation quantities *υ_i_*^−^(*l_k_*) at different stages of each level of the clearance angle in Table 12 were obtained.

According to Equations (11) and (12) and the ratio *r* = 1 between the total amount of excellent excitation and the total amount of bad excitation, the optimal excitation factor *h*^+^ = 0.47894 and the inferior excitation factor *h*^−^ = 0.52106 can be obtained.

According to Equation (10), the dynamic comprehensive evaluation value *z*(*t_k_*) of each level of the clearance angle in Table 13 at different stages is obtained. Set *τ_k_* = 1 and obtain the total dynamic comprehensive evaluation value *z* for each level of the rear Angle according to Equation (13). According to the total dynamic comprehensive evaluation value *z* of each level of the clearance angle in the last column in Table 13, 9° clearance angle is selected as the best level.

Similarly, the total dynamic comprehensive evaluation value of each level of spiral Angle, feed per tooth and cutting depth is shown in Table 14. According to this table, the optimal level of spiral Angle, feed per tooth and cutting depth is determined to be 30°, 0.15 mm/z, and 2.5 mm, respectively.

To sum up, when the rake angle is 8°, the cutting speed is 37.68 m/min, and the cutting width is 0.2 mm, the optimal combination of the four parameters studied is 9° clearance angle, 30° helix angle, 0.15 mm/z feed per tooth and 2.5 mm cutting depth.

### 3.6. Comparison between Parameter Combinations

After the optimal combination of tool parameters and cutting parameters was obtained by the dynamic evaluation method, the finite element method was used to DEFORM-3D finite element simulation of the cutting process of the group of parameters, and the tool wear rate values of the four stages under the group of parameters were obtained. According to Equation (2), the material removal rate of this group of parameters is obtained. Figure 9 is the simulation diagram of wear rate at each stage of parameter combination obtained by dynamic comprehensive evaluation method. Table 15 is the simulation value of tool wear rate at each stage and the numerical table of material removal rate.

By comparing Table 15 with Table 7, the tool wear rate values at each phase in Table 15 are better than most of the tool wear rate values at the same stage in Table 7, and the material removal rate in Table 15 is better than most of the material removal rate in Table 7. Therefore, the parameter combination obtained by dynamic evaluation method has higher comprehensive performance.

## 4. Validation of the Finite Element Model

In this chapter, the cutting force in three directions is taken as the performance indicator, and the cutting speed is taken as the research object. The accuracy of the simulation model is verified by comparing the simulation value with the experimental value. The reliability of the above research results can be illustrated by verifying the accuracy of the simulation model.

### 4.1. Setting of Experimental Parameters

Cutting speed is one of the important parameters affecting the cutting force [38]. Therefore, taking the cutting speed as the studied parameter, and taking the cutting forces in three directions as performance indicators, the accuracy of the finite element simulation model was verified by comparing the maximum values of the simulation values and the experimental values. Table 16 shows the milling parameters test table.

### 4.2. The Experimental Device

Figure 10 shows the end milling cutter, titanium alloy workpiece and instrument used in the experiment. Among them, Figure 10a is a CNC milling machine (VDL-1000E, Dalian Machine Tool Group, Dalian, China). Figure 10b is a rotary dynamometer (9171A, Kistler, Winterthur, Switzerland), an end mill and a titanium workpiece for measuring cutting forces in the x-, y-, and z-directions. The cutting force components along x-, y-, and z-directions are recorded by Dynoware signal analyzer software (Kistler, Winterthur, Switzerland).

### 4.3. Reliability Verification of Simulation Model

As shown in Table 17, this table is the cutting force value table under simulation and experiment. The experimental value is the cutting force value within a line, and the simulation value selects the cutting force data within a certain distance and deletes the unreasonable data. Figure 11 shows the comparison between the simulated and experimental values of the maximum cutting forces in three directions. According to the Figure 11, the variation trend of the simulation value and the experimental value with the cutting speed is basically the same, with the maximum error of 24.822% and the minimum error of 14.036%, so the simulation value is in good agreement with the experimental value, so the simulation model of the finite element model is reliable, the data obtained by finite element method can be used directly. Therefore, the reliability of the finite element simulation model shows that the dynamic evaluation results are also accurate and reliable.

## 5. Conclusions and Prospects

In order to solve the problems of short tool life and low machining efficiency in titanium alloy cutting process, it is necessary to study the optimal matching between tool parameters and cutting parameters. In this paper, the optimization of tool parameters and cutting parameters is studied. Taking titanium alloy side milling with end milling cutter as the research object, and taking tool wear rate and material removal rate as performance indicators. Clearance angle, helix angle, feed per tooth, and cutting depth are optimized parameters. The dynamic comprehensive evaluation method based on gain horizontal excitation is used to comprehensively evaluate each level of tool parameters and cutting parameters, and the optimal level of each parameter is obtained and combined to become the optimal parameter combination. After that, by comparing the optimized parameter combination with the previous parameter combination by finite element method, the comprehensive performance of the optimized parameter combination is determined to be higher. Finally, the reliability of the simulation model is verified by comparing simulation and experiment, so as to illustrate the reliability of the dynamic evaluation results. According to the above research, the following conclusions are drawn:In this paper, the dynamic evaluation method based on gain horizontal excitation was used to optimize the tool parameters and cutting parameters in the process of milling titanium alloy with milling cutter side, and the optimal matching combination of tool parameters and cutting parameters on the tool wear rate and material removal rate was obtained.When the rake angle is 8°, the cutting speed is 37.68 m/min, and the cutting width is 0.2 mm, the machining effect of the clearance angle is 9°, the helix angle is 30°, the feed per tooth is 0.15 mm/z, and the cutting depth is 2.5 mm achieves the best, which can simultaneously meet the requirements of long tool life and high machining efficiency. In addition, the reliability of simulation model is verified, and the optimization results are also reliable.The comparison between the optimized parameters by finite element method and the parameter combination in Table 6 shows that the optimized parameter combination has higher comprehensive performance.In this paper, the performance indicator value is obtained by simulation, but there is some error between simulation value and experimental value. Therefore, in the future, under the condition of sufficient time and funding, the required numerical value of tool wear rate and material removal rate will be obtained through experiments to make the optimization results more accurate.

## Figures and Tables

**Figure 1 materials-14-06181-f001:**
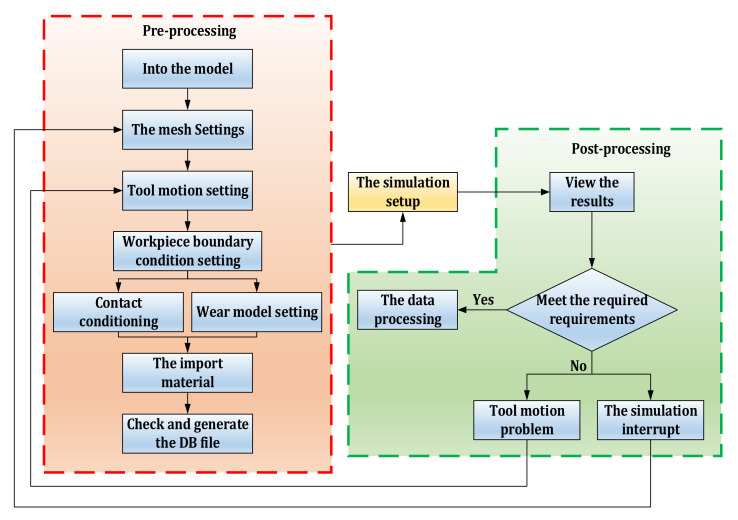
Cutting simulation model establishment process of DEFORM_3D software.

**Figure 2 materials-14-06181-f002:**
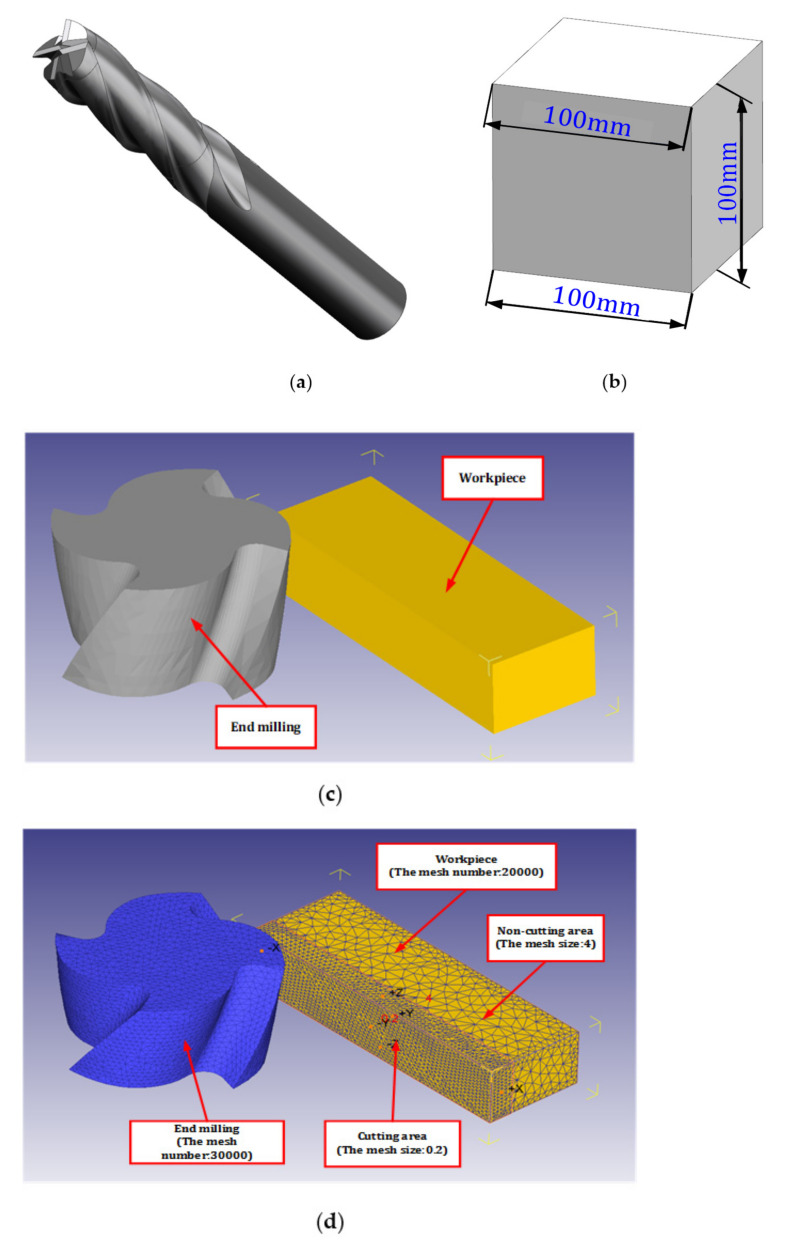
Simulation model of the side milling process: (**a**) End milling cutter model, (**b**) Workpiece model, (**c**) 3D models in simulation software and (**d**) The simulation model.

**Figure 3 materials-14-06181-f003:**
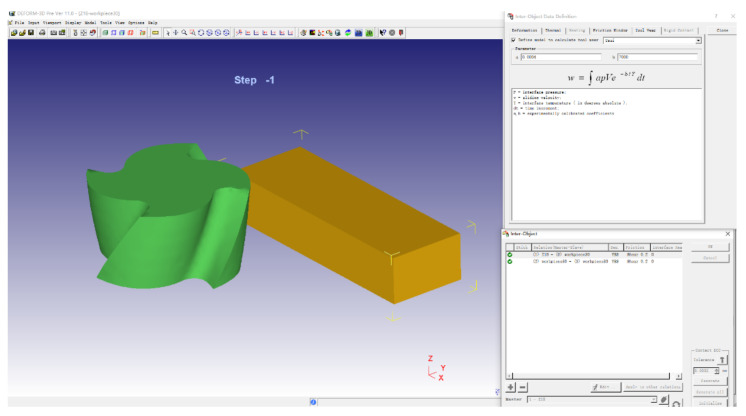
Tool wear model setting.

**Figure 4 materials-14-06181-f004:**
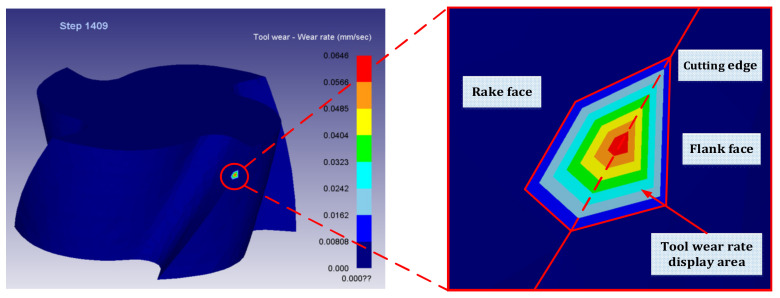
Simulation results of tool wear rate.

**Figure 5 materials-14-06181-f005:**
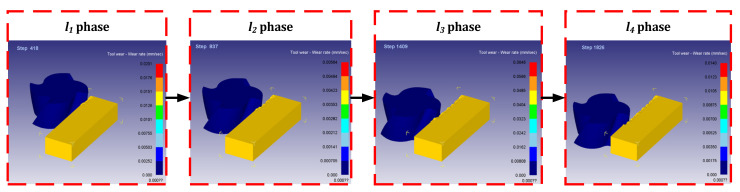
Dynamic change diagram of milling simulation.

**Figure 6 materials-14-06181-f006:**
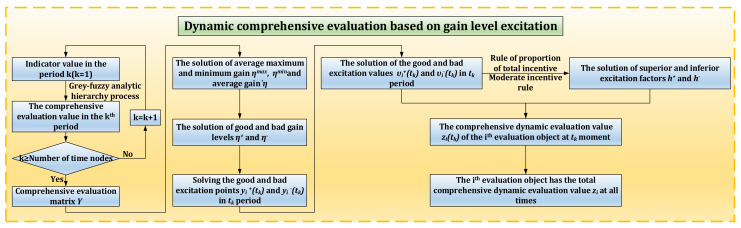
Flow chart of dynamic comprehensive evaluation method based on gain horizontal excitation.

**Figure 7 materials-14-06181-f007:**
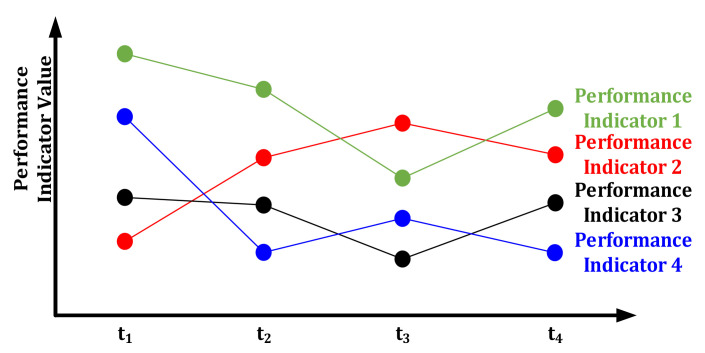
Dynamic change diagram of performance indicators.

**Figure 8 materials-14-06181-f008:**
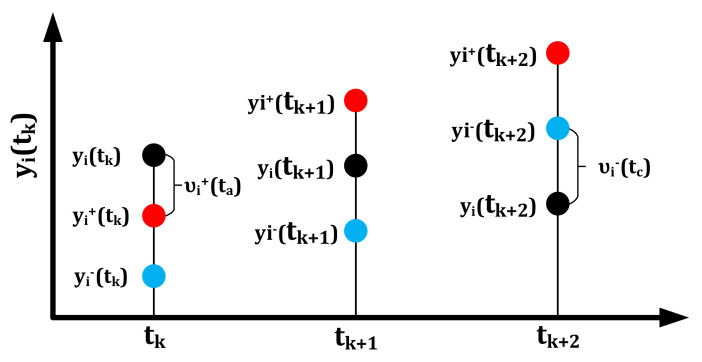
Geometric representation of excellent and bad excitation points and excitation quantities.

**Figure 9 materials-14-06181-f009:**
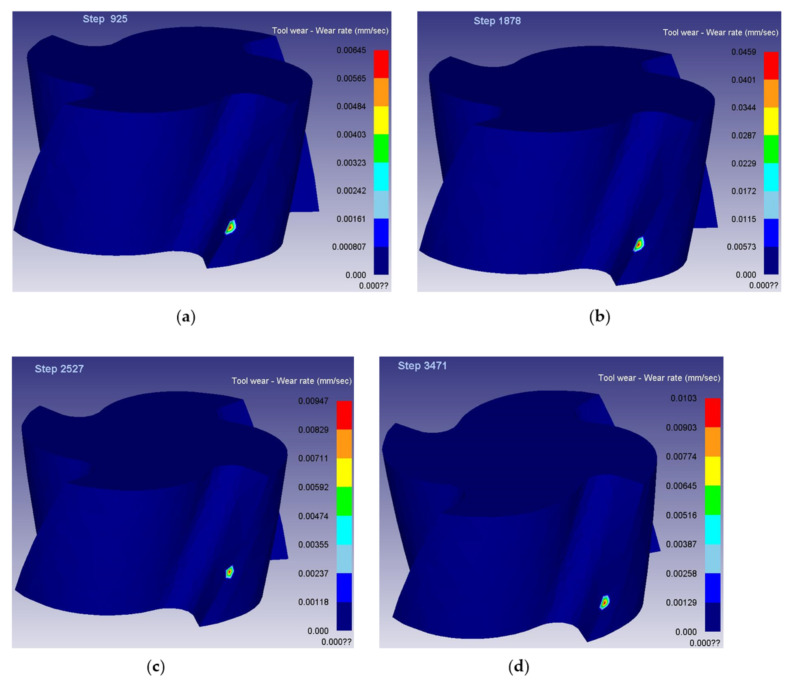
Simulation diagram of tool wear rate at each stage: (**a**) the first-stage wear rate value (**b**) the second-stage wear rate value (**c**) the third-stage wear rate value and (**d**) the fourth-stage wear rate value.

**Figure 10 materials-14-06181-f010:**
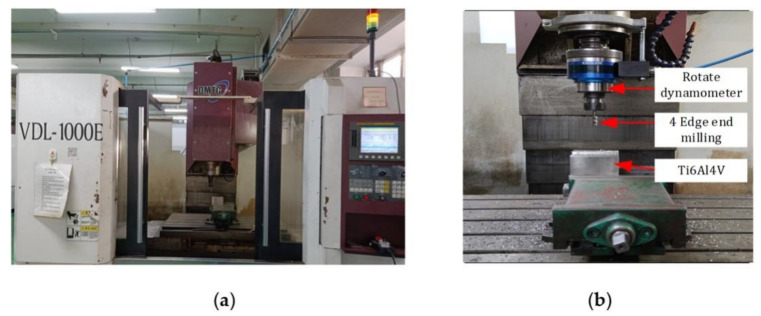
Experimental instruments and equipment used in the experiment (**a**) CNC milling machine and (**b**) cutting force measuring equipment.

**Figure 11 materials-14-06181-f011:**
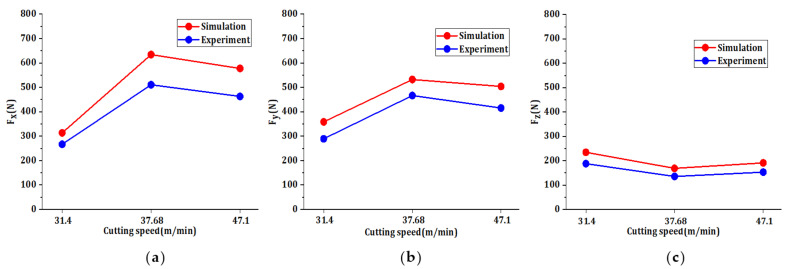
The simulation value of cutting force is compared with the experimental value: (**a**) x-direction (**b**) y-direction and (**c**) z-direction.

**Table 1 materials-14-06181-t001:** J-C parameters for Ti-6Al-4V alloy [27].

*A* (MPa)	*B* (MPa)	*C*	*m*	*n*	$\bar{\varepsilon_{0}}$ (s^−1^)	*T_m_* (°C)	*T_r_* (°C)
875	793	0.01	0.71	0.386	1	1560	20

**Table 2 materials-14-06181-t002:** Material parameter [28].

Material Parameter	YG6	Ti6Al4V
Density (g/cm^3^)	14.85	4.44
Young’s modulus (GPa)	640	112
Poission’s Ratio	0.22	0.34
Expansion (/°C)	4.7 × 10^−6^	9.4 × 10^−6^
Conductivity (W/m·K)	79.6	6.8
Specific heat (J/(kg·°C))	176	565

**Table 3 materials-14-06181-t003:** The key parameters of the integral end milling cutter.

Num.	Parameter	Value
1	The blade number	4
2	The cutter diameter	10 (mm)
3	The rake angle	8 (°)
4	The width of rake face	1.0 (mm)
5	The first clearance angle	12 (°)
6	The second clearance angle	23 (°)
7	The width of the first flank face	0.7 (mm)
8	The width of the second flank face	0.8 (mm)
9	The helix angle	35 (°)
10	The core diameter	6.2 (mm)

**Table 4 materials-14-06181-t004:** Table of fixed parameter values.

Rake Angle (°)	Cutting Speed (m/min)	Cutting Width (mm)
8	37.68	0.2

**Table 5 materials-14-06181-t005:** Variable parameter levels.

	Clearance Angle (°)	Helix Angle (°)	Feed Per Tooth (mm/z)	Cutting Depth (mm)
1	8.00	30.00	0.05	1.00
2	9.00	32.00	0.10	1.50
3	10.00	33.00	0.15	2.00
4	11.00	34.00	0.20	2.50
5	12.00	35.00	0.25	3.00

**Table 6 materials-14-06181-t006:** Orthogonal test table.

	Clearance Angle (°)	Helix Angle (°)	Feed Per Tooth (mm/z)	Cutting Depth (mm)
1	9.00	32.00	0.20	1.50
2	12.00	30.00	0.10	1.50
3	11.00	30.00	0.15	2.00
4	8.00	32.00	0.25	2.00
5	10.00	30.00	0.20	2.50
6	9.00	33.00	0.15	2.50
7	11.00	32.00	0.10	3.00
8	12.00	32.00	0.05	2.50
9	12.00	34.00	0.20	2.00
10	11.00	34.00	0.25	2.50
11	10.00	34.00	0.05	3.00
12	9.00	34.00	0.10	1.00
13	9.00	30.00	0.25	3.00
14	11.00	33.00	0.05	1.50
15	8.00	33.00	0.20	3.00
16	8.00	30.00	0.05	1.00
17	10.00	32.00	0.15	1.00
18	12.00	35.00	0.15	3.00
19	8.00	34.00	0.15	1.50
20	10.00	35.00	0.25	1.50
21	9.00	35.00	0.05	2.00
22	11.00	35.00	0.20	1.00
23	12.00	33.00	0.25	1.00
24	8.00	35.00	0.10	2.50
25	10.00	33.00	0.10	2.00

**Table 7 materials-14-06181-t007:** Performance indicator data table.

	The First Stage (*l*_1_)		The Second Stage (*l*_2_)		The Third Stage (*l*_3_)		The Fourth Stage (*l*_4_)	
	Wear Rate (mm/s)	*V* (mm^3^/s)	Wear Rate (mm/s)	*V* (mm^3^/s)	Wear Rate (mm/s)	*V* (mm^3^/s)	Wear Rate (mm/s)	*V* (mm^3^/s)
1	0.00934	4.8	0.00547	4.8	0.00156	4.8	0.02260	4.8
2	0.07030	2.4	0.04810	2.4	0.06770	2.4	0.13100	2.4
3	0.01680	4.8	0.01240	4.8	0.00945	4.8	0.02470	4.8
4	0.00623	8.0	0.01920	8.0	0.01110	8.0	0.13000	8.0
5	0.03900	8.0	0.14100	8.0	0.03130	8.0	0.02760	8.0
6	0.00718	6.0	0.03550	6.0	0.00378	6.0	0.01050	6.0
7	0.00168	4.8	0.00718	4.8	0.11400	4.8	0.01590	4.8
8	0.02320	2.0	0.01390	2.0	0.01760	2.0	0.01180	2.0
9	0.00508	6.4	0.03820	6.4	0.04800	6.4	0.10500	6.4
10	0.01030	10.0	0.02150	10.0	0.00221	10.0	0.00834	10.0
11	0.00338	2.4	0.22500	2.4	0.00802	2.4	0.05170	2.4
12	0.06360	1.6	0.11400	1.6	0.04850	1.6	0.05840	1.6
13	0.00747	12.0	0.05300	12.0	0.09910	12.0	0.06970	12.0
14	0.01660	1.2	0.04650	1.2	0.01180	1.2	0.07540	1.2
15	0.05810	9.6	0.02610	9.6	0.05140	9.6	0.01940	9.6
16	0.01460	0.8	0.03630	0.8	0.07520	0.8	0.04940	0.8
17	0.00944	2.4	0.03190	2.4	0.07750	2.4	0.00726	2.4
18	0.02010	6.4	0.00564	6.4	0.06460	6.4	0.01400	6.4
19	0.00544	3.2	0.00826	3.2	0.00757	3.2	0.01910	3.2
20	0.03630	6.0	0.09990	6.0	0.03680	6.0	0.03020	6.0
21	0.01550	1.6	0.00282	1.6	0.01850	1.6	0.00993	1.6
22	0.01930	3.2	0.01340	3.2	0.02070	3.2	0.01630	3.2
23	0.01380	4.0	0.03740	4.0	0.01010	4.0	0.00943	4.0
24	0.04980	4.0	0.01570	4.0	0.02950	4.0	0.03220	4.0
25	0.01970	3.2	0.01060	3.2	0.04600	3.2	0.02340	3.2

**Table 8 materials-14-06181-t008:** Time series stereoscopic data table.

	*t* _1_				*t* _2_				…	*t_T_*			
	*x* _1_	*x* _2_	…	*x_m_*	*x* _1_	*x* _2_	…	*x_m_*	…	*x* _1_	*x* _2_	…	*x_m_*
*s* _1_	*x*_11_(*t*_1_)	*x*_12_(*t*_1_)	…	*x*_1*m*_(*t*_1_)	*x*_11_(*t*_2_)	*x*_12_(*t*_2_)	…	*x*_1*m*_(*t*_2_)	…	*x*_11_(*t_T_*)	*x*_12_(*t_T_*)	…	*x*_1*m*_(*t_T_*)
*s* _2_	*x*_21_(*t*_1_)	*x*_22_(*t*_1_)	…	*x*_2*m*_(*t*_1_)	*x*_21_(*t*_2_)	*x*_22_(*t*_2_)	…	*x*_2*m*_(*t*_2_)	…	*x*_21_(*t_T_*)	*x*_22_(*t_T_*)	…	*x*_2*m*_(*t_T_*)
…	…	…	…	…	…	…	…	…	…	…	…	…	…
*s_n_*	*x*_*n*1_(*t*_1_)	*x*_*n*2_(*t*_1_)	…	*x_nm_*(*t*_1_)	*x*_*n*1_(*T*_2_)	*x*_*n*2_(*t*_2_)	…	*x_nm_*(*t*_2_)	…	*x*_*n*1_(*t_T_*)	*x*_*n*2_(*t_T_*)	…	*x_nm_*(*t_T_*)

**Table 9 materials-14-06181-t009:** Significance comparison scale values among indicators.

*x_j_*/*x_h_*	*x*_1_/*x*_2_	*x*_2_/*x*_2_
*U_jh_*	1.5	1.0

**Table 10 materials-14-06181-t010:** Stage sequence stereoscopic data table of the clearance angle.

	The First Stage (*l*_1_)		The Second Stage (*l*_2_)		The Third Stage (*l*_3_)		The Fourth Stage (*l*_4_)	
	Wear Rate (mm/s)	*V* (mm^3^/s)	Wear Rate (mm/s)	*V* (mm^3^/s)	Wear Rate (mm/s)	*V* (mm^3^/s)	Wear Rate (mm/s)	*V* (mm^3^/s)
1	0.02683	5.12	0.02111	5.12	0.03495	5.12	0.05002	5.12
2	0.02062	5.20	0.04216	5.20	0.03429	5.20	0.03427	5.20
3	0.09176	4.40	0.10168	4.40	0.03992	4.40	0.02803	4.40
4	0.01150	5.04	0.01728	5.04	0.04477	5.04	0.01450	5.04
5	0.02650	4.24	0.02865	4.24	0.04160	4.24	0.26369	4.24

**Table 11 materials-14-06181-t011:** The excellent and bad excitation points of each level of the clearance angle at different stages.

	*l* _1_	*l* _2_		*l* _3_		*l* _4_	
	*y_i_*(*l*_1_)	*y_i_*^+^(*l*_2_)	*y_i_*^−^(*l*_2_)	*y_i_*^+^(*l*_3_)	*y_i_*^−^(*l*_3_)	*y_i_*^+^(*l*_4_)	*y_i_*^−^(*l*_4_)
1	0.77701	0.76937	0.74242	0.88793	0.86098	0.86810	0.84115
2	0.88889	0.88125	0.85430	0.77800	0.75105	0.99236	0.96541
3	0.75000	0.74236	0.71541	0.74236	0.71541	0.43160	0.40465
4	0.90000	0.89236	0.86541	0.89236	0.86541	0.49236	0.46541
5	0.57008	0.56244	0.53549	0.60416	0.57721	0.37621	0.34926

**Table 12 materials-14-06181-t012:** Good and bad excitation scales for each level of clearance angle at different stages.

	*l* _1_		*l* _2_		*l* _3_		*l* _4_	
	*υ_i_*^+^(*l*_1_)	*υ_i_*^−^(*l*_1_)	*υ_i_*^+^(*l*_2_)	*υ_i_*^−^(*l*_2_)	*υ_i_*^+^(*l*_3_)	*υ_i_*^−^(*l*_3_)	*υ_i_*^+^(*l*_4_)	*υ_i_*^−^(*l*_4_)
1	0	0	0.12640	0	0	0	0	0.03139
2	0	0	0	0.06866	0.22800	0	0	0.04758
3	0	0	0.00764	0	0	0.27617	0.25963	0
4	0	0	0.00764	0	0	0.36541	0.40764	0
5	0	0	0.04936	0	0	0.19336	0	0.01593

**Table 13 materials-14-06181-t013:** The dynamic comprehensive evaluation table of each level of the clearance angle and the total dynamic comprehensive evaluation table.

	*z*(*l*_1_)	*z*(*l*_2_)	*z*(*l*_3_)	*z*(*l*_4_)	*z*
1	0.77701	0.95611	0.87574	0.79340	3.40295
2	0.88889	0.74986	1.10920	0.89304	3.65099
3	0.75000	0.75366	0.29534	0.70366	2.50266
4	0.9000	0.90366	0.30960	1.09524	3.20850
5	0.57008	0.63544	0.28310	0.32503	1.81365

**Table 14 materials-14-06181-t014:** Total dynamic composite evaluation values for each level of the remaining parameters.

	*z* * _Helix angle_ *	*z* * _Feed per tooth_ *	*z* * _Cutting depth_ *
1	2.44285	1.99773	1.75018
2	2.20450	1.62204	1.77147
3	2.31967	3.05105	2.48153
4	2.04067	2.39639	2.93874
5	2.06289	2.95389	2.67173

**Table 15 materials-14-06181-t015:** The numerical table of tool wear rate and material removal rate corresponding to the optimal parameters.

Wear Rate (mm/s)				*V* (mm^3^/s)
*l* _1_	*l* _2_	*l* _3_	*l* _4_	
0.00645	0.0459	0.00947	0.0103	6.0

**Table 16 materials-14-06181-t016:** Settings of milling cutting parameters.

Cutting Speed (m/min)	Feed Speed (mm/min)	Cutting Depth (mm)	Cutting Width (mm)
31.40	400	3	0.8
37.68	400	3	0.8
47.10	400	3	0.8

**Table 17 materials-14-06181-t017:** Obtained cutting forces for different cutting speeds.

	Cutting Speed (m/min)	Simulation (N)	Experiment (N)
*F_x_*	31.40	313.613	266.791
	37.68	634.243	510.750
	47.10	577.552	462.902
*F_y_*	31.40	358.606	289.408
	37.68	532.094	466.600
	47.10	503.908	415.483
*F_z_*	31.40	234.752	188.069
	37.68	168.884	135.653
	47.10	191.216	153.299

## Data Availability

Not applicable.

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
