# Peer review of "Optimization Method of Tool Parameters and Cutting Parameters Considering Dynamic Change of Performance Indicators"

_materials, 2021, doi:10.3390/ma14206181_

Round 1

Reviewer 1 Report

I have reviewed the paper “Optimization Method of Tool Parameters and Cutting Parameters Considering Dynamic Change of Performance Indicators” and I would like to congratulate the authors for their work. In general, the work is interesting, but I have several major concerns that need to be addressed from the authors. Therefore, I would like from the authors to elaborate the next comments:

  1. Please revise the next long sentence in the abstract of the paper:

With high efficiency and long tool life, for the purpose of tool wear rate and material removal rate as performance indicators, in front of the rake angle, cutting speed, cutting width of vertical milling cutter side milling titanium alloy as the research object, using the dynamic evaluation method of the tool parameters and cutting parameters in milling process optimization, finally got to match, the parameter of the performance indicators to achieve the best combination’.

  1. The literature review should include recent studies concerning Machine Learning approaches in estimating cutting forces such as:

Alajmi, M.S.; Almeshal, A.M., Modeling of Cutting Force in the Turning of AISI 4340 Using Gaussian Process Regression Algorithm. Applied Sciences 2021, 11, 4055

Charalampous P., Prediction of Cutting Forces in Milling Using Machine Learning Algorithms and Finite Element Analysis, Journal of Materials Engineering and Performance 2021, Vol. 30, Pages 2002-2013

B.Peng, T. Bergs, D.Schraknepper, F. Klocke, B. Döbbeler, A hybrid approach using machine learning to predict the cutting forces under consideration of the tool wear, Procedia CIRP, 2019, Vol. 82, Pages 302-307

  1. The novelty of this research should be included in the introduction section of the manuscript with more details.
  2. Please revise the next sentence in page 3 of the paper:

‘Using this software to simulate metal cutting process on computer can reduce The times of cutting tool test and reduce the cost of material and energy consumption’

  1. Is the number of elements in the workpiece material enough for your FEA? Did you conduct FE simulations using other total number of elements?
  2. Please include data about the cutting tool, such us the type of cutting insert, material of the substrate and the potential coating and finally the geometrical characteristics.
  3. I do not think the numbering (1, 2, 3, ...9) in Section 3.3 Parameter level optimization based on dynamic evaluation method and in Section 3.4 Comprehensive evaluation of parameter level in each stage (1, 2, 3, ...7)  is necessary.
  4. Can your methodology be employed in using other values in crucial parameters like the cutting speed (you used a constant 37.68 m/min)?
  5. Additional experimental procedures are necessary in order to validate the results of Section 3.
  6. How could your work be applied in industry-wise applications in order to enhance the manufacturing process of milling?
  7. English should be rechecked and improved for better comprehension.

Reviewer 2 Report

The authors of the manuscript presented a method of optimizing selected geometric parameters of the tool and selected cutting parameters. The goal of optimization is to extend tool life while maximizing machining efficiency. In order to improve the quality of the manuscript, I propose to consider my following comments:

1. In the first chapter, the fragment concerning the analysis of work 3 does not correspond to the subject of the manuscript.

2. The authors inform that the geometric models of the tool and the workpiece were simplified. It is not known what these simplifications were. Were there any simplifications related to the analyzed geometric parameters of the tool?

3. Figures 3a and 3b show only the standard windows of the used software. 

4. Table 6 - The tool wear is not taken into account when calculating the machining efficiency, it is unnecessary to repeat the same values of the material removal rate for each periods. In the first roow the value of the ware rate for period three is incorrect. 

5. FEM calculations of the tool wear are verified only based on the cutting force analysis.  In my opinion, such verification is unnecessary. Why the calculations were not verified on the basis of the tool wear observed during the conducted experiment. 

Reviewer 3 Report

The optimization of the metal cutting process, including the tool geometrical parameters and the machining parameters, is of interest to manufacturing engineers, as it conducts to extended tool life, high productivity, and high surface quality of the workpiece. 

Regarding the paper entitled "Optimization Method of Tool Parameters and Cutting Parameters Considering Dynamic Change of Performance Indicators", in the current stage its originality is low, as it uses the dynamic evaluation method based on gain horizontal excitation already presented in [14]. In my opinion, the presentation of the optimization method could be condensed at its essentials.

Also, there is no graphical representation of the tool wear and cutting efficiency versus the optimization parameters.

Even if the poor experimental part shows that the trend of the measured cutting forces is the same as simulated, more experiments are necessary. This could be the strong point of the paper in the future, bringing some originality.

Furthermore, the paper is written in an alembicated way, mixing parts that should be transferred to introduction sections with repeated phrases on the same ideas. The authors acknowledge the help of English proficients, but the paper still suffers from this viewpoint, e.g., the second phase of the first paragraph from the conclusions section is nonsense.

I advise the authors to carry out more experiments and to improve their model as planned, and resubmit the paper after considering all the above recommendations. 

Round 2

Reviewer 1 Report

The paper “Optimization Method of Tool Parameters and Cutting Parameters Considering Dynamic Change of Performance Indicators” has significantly improved its quality. The authors of the study have properly answered all of my questions and comments. Hence, the manuscript can be accepted for publication in its present form. Finally, I would like to thank the authors for their effort to response my comments and questions.

Reviewer 2 Report

The authors of the manuscript  made the necessary improvements. In my opinion, the manuscript may be published.
Before publication, please check the Figure 2b - in my opinion, one of the dimensions is incorrectly marked.

Reviewer 3 Report

At a first glance, I see in the revised version of the article new incomplete phrases, e.g., in the abstract:

"For the purpose of high processing efficiency and long tool life, with tool wear rate and material removal rate as performance indicators."

In the Introduction:

"Kubilay et al. [3] taking the turning of Ti6Al4V with indexable turning tool as the research object."

"Viswanathan et al. [10] taking PVD coated carbide turning tool dry turning magnesium alloy as the research object, taking cutting speed, feed per revolution and cutting depth as the optimized parameters, and taking cutting force, material removal rate, flank face wear and surface roughness as the performance indicators."

"Carry out parameter optimization test. "...

in the Conclusions section: 

"In this paper, the optimization of tool parameters and cutting parameters is studied. Taking titanium alloy side milling with end milling cutter as the research object, and taking tool wear rate and material removal rate as performance indicators. "

and so on all over the amended part of the article.

Anyway, the paper was much improved, presenting new FEM simulation results on the milling process. 

For this reason, the paper has my acceptance, as the language mistakes are not the business of the reviewers.